

# Unexpected scarcity of ANME Archaea in hydrocarbon seeps within Monterey Bay

Amanda C. Semler[1], Anne E. Dekas[1]

[1]Department of Earth System Science, Stanford University, Stanford, 94305, USA

*Correspondence to*: Amanda C. Semler (semler@stanford.edu) and Anne E. Dekas (dekas@stanford.edu)

**Abstract.** Marine hydrocarbon seeps typically harbor a relatively predictable microbiome, including anaerobic methanotrophic (ANME) archaea. Here, we sampled two cold seeps in Monterey Bay, CA – Clam Field and Extrovert Cliff – which have been known for decades but never characterized microbiologically. Many aspects of these seeps were typical of seeps worldwide, including elevated methane and sulfide concentrations, $^{13}$C-depleted

dissolved inorganic carbon, and the presence of characteristic macrofauna. However, we observed atypical microbial communities: extremely few ANME sequences were detected in either 16S rRNA or *mcrA* gene surveys at Clam Field (<0.1% of total community reads), even after six months of incubation with methane in the laboratory, and only slightly more ANME sequences were recovered from Extrovert Cliff (<0.3% of total community reads). At Clam Field, a lack of ANME *mcrA* transcription, a lack of methane-dependent sulfate

reduction, and a linear porewater methane profile were consistent with low or absent methanotrophy. Although the reason for the scarcity of ANME is yet unclear, we postulate that non-methane hydrocarbon release excludes anaerobic methanotrophs directly or indirectly (e.g., through competitive interactions with hydrocarbon-degrading bacteria). Our findings highlight the potential for hydrocarbon seeps without this critical biofilter, with implications for their contribution to global methane emissions.

## 20   1 Introduction

Monterey Bay is a well-studied region of the California coastline located within a tectonically active transform boundary. Right-lateral, strike-slip motion between the Pacific Plate to the west and the North American Plate to the east produces movement in the bay along two major fault zones (the San Gregorio and Monterey Bay fault zones) and causes extensive sedimentary compression and compaction (Clark, 1981; Orange et al., 1999).

These forces drive fluid flow through the organic-rich, hydrocarbon-bearing sediments underlying Monterey Bay (Orange et al., 1999; Stakes et al., 1999), creating large networks of cold seeps – highly productive chemosynthetic



ecosystems on the seafloor typically characterized by methane- and sulfide-rich fluids. In Monterey Bay, cold seeps are concentrated especially around high-porosity sediment layers and permeable fractures within the fault zones (Moore et al., 1991; Greene et al., 1999; Orange et al., 1999), as well as sites of recent erosion such as

canyon walls (Paull et al., 2005).

        Cold seeps have been surveyed and studied in Monterey Bay for more than three decades (Barry et al., 1996; Orange et al., 1999; Lorenson et al., 2002) – almost from the time cold seeps were first discovered (Paull et al., 1984; Suess et al., 1985). Investigations of Monterey Bay seeps have been particularly focused on their fluid chemistry and macrofaunal communities. Previous chemical analyses have demonstrated that fluids at most

Monterey Bay seeps are enriched in sulfide and methane, and the $^{13}$C-depleted dissolved inorganic carbon (DIC) in pore fluids and in authigenic carbonates surrounding the seeps suggest the original methane is of a largely microbial origin (Martin et al., 1997). However, methane at some sites also has a distinct thermogenic isotope imprint, with potential input from deep fluids flowing through the organic-rich Monterey Formation (Martin et al., 1997; Rathbun et al., 2003; Füri et al., 2009). Non-methane hydrocarbons – including ethane, propane, and

butane, as well as visible oil – have also been discovered in the seep fluids (Lorenson et al., 2002), especially at seeps located within the Monterey Bay Fault Zone. Utilizing the reduced compounds in Monterey Bay seep fluids (either directly, or indirectly through a symbiont) are high numbers of vesicomyid clams, thiotrophic Beggiatoa mats, and, more rarely, vestimentiferan and pogonophoran tube worms (Fisher and Childress, 1992; Greene et al., 1994; Orange et al., 1994; Barry et al., 1996, 1997) – an overall community which, at broad, family-level

taxonomic scales, mirrors the macrofauna found at cold seeps across the Pacific Basin (Barry et al., 1996). Species-level differences between macrofaunal taxa in Monterey Bay have been largely attributed to the differing sulfide concentrations at each individual seep (Barry et al., 1996).

        However, while much is known about the geology, geochemistry, and macrofauna of Monterey Bay cold seeps, few investigations have targeted microbial communities here. Seep sediment has been previously retrieved

from one Monterey Bay seep – Extrovert Cliff – for incubations in bioreactors (Girguis et al., 2005, 2003), and archaeal-specific 16S rRNA primers were utilized to generate clone libraries before and after enrichment. Clones belonging to characteristic seep microbial taxa were recovered – specifically, clones of anaerobic methanotrophic (ANME) archaea subgroups ANME-2b and ANME-2c. ANME archaea are core microbial taxa at cold seeps, as they couple the anaerobic oxidation of methane (AOM) to sulfate reduction with the help of syntrophic sulfate-

reducing bacterial (SRB) partners (Boetius et al., 2000; Orphan et al., 2001). ANME are comprised of three distinct polyphyletic clusters in the phylum Halobacterota (ANME-1, ANME-2, and ANME-3), and they associate and share electrons with a variety of SRB partners (Seep-SRB1, Seep-SRB2, Seep-SRB4, and thermophilic Hot-





Seep1) typically in tight cell aggregates (in the case of ANME-2 and ANME-3). Both ANME and their symbionts are key components of the "seep microbiome" (Ruff et al., 2015) – the core set of microbial groups that dominate

seeps globally. However, it has been noted that quantitative surveys for ANME archaea using fluorescence *in-situ* hybridization (FISH) have been largely unsuccessful in Monterey Bay sediments, and cells with the typical ANME aggregate morphology are rare, even at Extrovert Cliff where ANME-2 clones were recovered (Girguis et al., 2003).

Here, we investigated the microbial communities and sediment porewater geochemistry at two cold seeps

in Monterey Bay: Clam Field (895-909 mbsl) and Extrovert Cliff (965-990 mbsl). Using deep amplicon sequencing of both the 16S rRNA and methyl coenzyme-M reductase (*mcrA*) genes, we characterized archaeal and bacterial community composition (DNA) and potential activity (RNA) in 20-cm sediment cores collected along 100m transects from the center of each seep to "background" sediment. Using droplet digital PCR (ddPCR), we also investigated the abundance of *mcrA* genes and transcripts inside and outside each seep. Sediment from

the Clam Field site was also collected and incubated under a variety of methane headspace concentrations for six months to enrich methanotrophic taxa and their sulfate-reducing symbionts. Our goals were to: i) characterize the community of microorganisms within these well-known cold seeps; and ii) evaluate the impact of Monterey Bay's unusual and complex hydrocarbon geochemistry on seep communities. This comparative analysis provides the first deep-sequencing perspective of Monterey Bay cold seeps, and, more broadly, helps us understand how local

geochemistry impacts methane oxidation potential.

## 2 Materials and Methods

### 2.1 Site Description and Geochemical Conditions

The two Monterey Bay cold seep sites investigated in this study were Clam Field and Extrovert Cliff (Fig. 1a). The Clam Field site is located on a sedimented apron of the Monterey Canyon wall and is constituted

by a broad band of seepage parallel to the canyon. Dense fields of live clams are found across the main band of seepage, along with patchy microbial mats (Fig. 1b-c). Clam Field is located within the Monterey Bay fault zone (Orange et al., 1999), suggesting that fluid seepage there is tectonically influenced and likely driven by artesian flow through the highly fractured, hydrocarbon-rich shale of the Monterey Formation (Barry et al., 1996; Lorenson et al., 2002; LaBonte et al., 2007; Füri et al., 2009). Previously measured $\delta^{13}C$ values of methane at this site (-50

to -55‰) are indicative of a high degree of thermogenic input – more so than at many other Monterey Bay seeps (Lorenson et al., 2002). The Extrovert Cliff site is located on the slope of a slide scar between the San Gregorio





and Monterey Bay fault zones. The site is characterized by distinct, concentric rings of seepage covered by thick microbial mats and bordered by live clams (Fig. 1d-e). Fluid flow rates at Extrovert Cliff are temporally variable and tidally influenced, suggesting fluid conduits from an overpressurized aquifer (LaBonte et al., 2007; Füri et al., 2009).


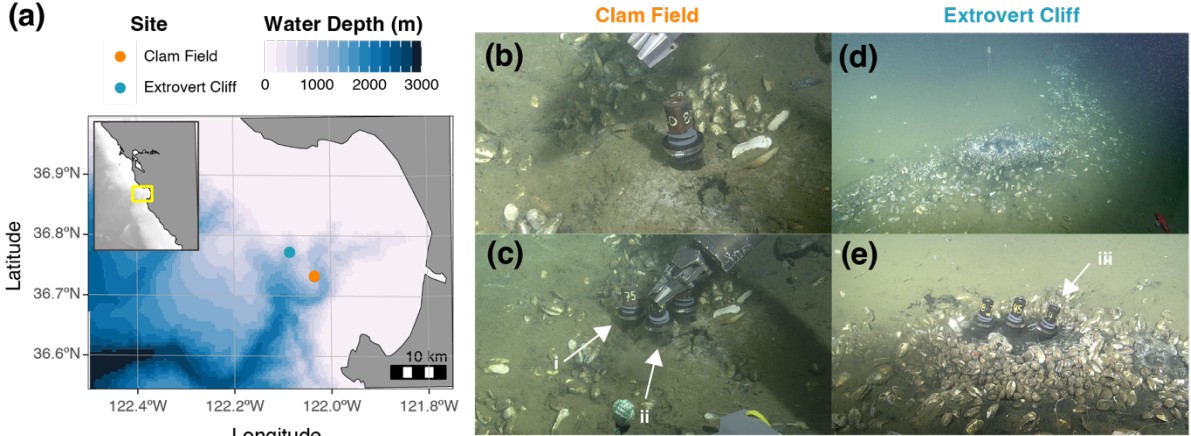

**Fig. 1: (a) Bathymetric map of Monterey Bay and the location of the two sampling sites: Clam Field (CF – 895-909 mbsl), and Extrovert Cliff (EC – 965-990 mbsl). Inset shows the central California coast, with Monterey Bay indicated by a yellow box. ROV Doc Ricketts images of seep surface expression before (b) and during (c) sampling at Clam Field, and before (d) and during (e) sampling at Extrovert Cliff. Pushcores collected for in-situ measurements (i; PC 75) and incubations (ii; PC 54) at the Clam Field "Seep" location, and for in-situ measurements at the Extrovert Cliff "Seep" location (iii; PC 64), indicated by white arrows.**


### 2.2 Sample collection and processing


Sediment pushcores were collected from both seep sites in April 2019 on the R/V *Western Flyer*, using ROV *Doc Ricketts*. At each site, two pushcores up to 20 cm long were collected from each of four locations: 1) the center of the cold seep, 2) the inner edge of the cold seep, 3) 5 meters outside the seep boundary, and 4) 100 meters outside the seep boundary (Fig. 2a). These were categorized as "Seep," "Seep-Edge", "Background-5m," or "Background-100m" cores, respectively (Table S1). Seep boundaries were delineated by the sudden termination of white, filamentous microbial mats and live clam beds on the sediment surface. These boundaries were validated by the sulfidic smell of the Seep and Seep-Edge cores from each site once cores were recovered.






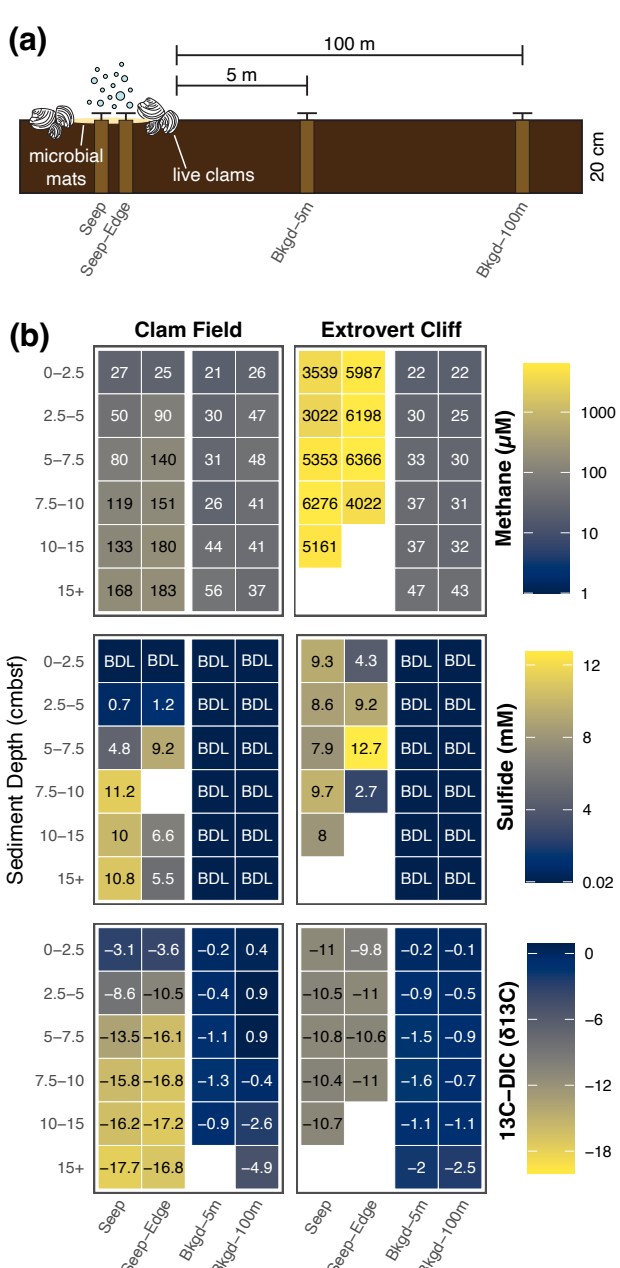

Fig. 2: (a) Diagram (not to scale) of core sampling scheme at both Clam Field and Extrovert Cliff. (b) Methane and sulfide concentration and δ13C-DIC values with sediment depth in all sampled cores. Blank



**values indicate no measurement was made. BDL indicates a value below the detection limit of the assay; detection limits were 1 μM and 0.02 mM for methane and sulfide, respectively.**

Onboard, cores were kept at 4 ºC until extruded from pushcore liners and sectioned within 8 hours of
collection. One core of each pair was sectioned into 2.5 cm (0-10 cmbsf) or 5 cm (10+ cmbsf) horizons and subsampled for molecular and geochemical analyses, while the other was sectioned into 5 cm horizons and preserved anaerobically within Whirlpaks sealed in mylar bags at 4 ºC for incubation experiments. Several 1 mL subsamples of each depth horizon in the molecular/geochemical cores were immediately flash-frozen in liquid nitrogen and preserved at -80 ºC for later DNA and RNA extraction. Subsamples of 3 mL were transferred into
25 mL butyl-rubber sealed vials filled with 5 mL of 5M sodium hydroxide solution for methane analysis, 3-5 mL subsamples were scooped onto a pre-weighed sheet of aluminum foil for dehydration and porosity measurements, and 0.5 mL subsamples were transferred into 2 mL epitubes pre-loaded with 1 mL of 4% PFA for microscopy. Porewater was squeezed from sediments using a porewater pressing bench (KC Denmark Research Equipment, Silkeborg, Denmark) under a stream of argon gas immediately after sectioning, and porewater was filtered with
0.2 μm Durapore® PVDF membrane filters (EMD Millipore, Burlington, MA, USA). 0.5 mL porewater was fixed with 0.5 mL of 0.5M zinc acetate and stored at 4 ºC for sulfide measurements, and 2 mL porewater was added to a 12 mL Exetainer® (LabCo Limited, Ceredigion, UK) pre-loaded with 1 mL 85% phosphoric acid, evacuated, and $N_2$-flushed for $\delta^{13}C$ analysis of dissolved inorganic carbon (DIC). The remaining porewater was stored at -20 ºC.

**2.3 Incubation setup**

Preserved sediment from all four Clam Field locations was anaerobically incubated and subsampled according to the method described in Dekas et al. (2009, 2014, 2016). The top two horizons (0-5 and 5-10 cm) from the "Seep" and "Seep-Edge" cores, and the top three horizons (0-5, 5-10, and 10-15 cm) from the "Background-5m" and "Background-100m" cores ($n_{horizons}$ = 10) were incubated with varying concentrations (0 –
2 atm) of methane to evaluate community responses to methane addition; $^{15}NH_4^+$ was also added to all incubations at a final concentration of 100 μM to measure total anabolic activity (99 atom % $^{15}N$; Cambridge Isotopes, NLM-467-1). The methane was a 4:1 mixture of natural abundance $^{13}C/^{12}C$ and 99 atom % $^{13}C$ methane (Sigma-Aldrich, 490229-1L). Table S2 lists each of five methane treatments per horizon; all treatments were performed in triplicate ($n_{incubations}$ = 150). Incubations were subsampled at 0, 0.5, 1, 3, and 6 month timepoints.





**2.4 Porewater geochemistry**


Headspace methane concentrations were measured from butyl-rubber sealed vials (see above) using gas chromatography coupled to flame-ionization detection (Shimadzu GC-2014, Stanford University, Stanford, CA, USA), and back-calculated to porewater concentrations using porosity values determined via weight loss after dehydration (and assuming a sediment density of 2.65 g cm$^{-3}$). Sulfide concentrations were measured in triplicate

and determined colorimetrically from the zinc acetate-preserved porewater samples using the methylene blue method (Cline, 1969), with a detection limit of 0.02 mM. All other assays were performed without replication due to limitations on porewater volume.

**2.5 Nucleic acid extraction, amplification, and sequencing of 16S rRNA and *mcrA* genes and transcripts**

DNA and RNA were extracted from flash-frozen sediments using the RNeasy Powersoil Total RNA
Isolation Kit (RNA) and the RNA Powersoil DNA Elution Accessory Kit (DNA; MoBio Laboratories, Carlsbad, CA, USA). The protocol was modified from the manufacturer's instructions to include a bead beating step of 60 s at a speed of 5.5 m/s on a FastPrep instrument (MP Biomedicals, Santa Ana, CA, USA) to facilitate archaeal cell lysis as in Semler et al. (2022). RNA extracts were cleaned with the Ambion TURBO DNA-free Kit (ThermoFisher Scientific, Waltham, MA, USA), and reverse transcription of RNA to cDNA was completed using
Superscript III First Strand Synthesis Supermix (Thermofisher Scientific, Waltham, MA, USA).

DNA and cDNA were concentration normalized and amplified using a two-step PCR plan for Illumina amplicon sequencing. In the first step, universal primers 515F-Y/926R (Parada et al., 2016) were used to target the V4-V5 region of the 16S rRNA gene sequence, and mcrA_F/mcrA_R (Luton et al., 2002; Dekas et al., 2016) was used to target *mcrA* genes and transcripts. Both sets of primers included an extension complimentary to the
primers used in the second PCR. The gene-targeting region of the primer sequences are listed in Table S3. 25 µL PCR reactions were performed containing 0.5 µL of forward and 0.5 µL of reverse primers (10 µM concentration), 10 µL 5PRIME HotMasterMix (2.5x, Quanta-Bio, Beverly, MA, USA), 13 µL DNase-free water, and 1 µL DNA or cDNA template. For 515F-Y/926R primers, the thermal cycling conditions were as follows: initial denaturing at 95 ℃ for 180 s; 28 cycles of 95 ℃ for 45 s, 50 ℃ for 45 s, and 68 ℃ for 90 s; a final elongation step at 68 ℃
for 300 s; and refrigeration at 4 ℃ until removal and storage. For mcrA_F/mcrA_R primers, the thermal cycling conditions were as follows: initial denaturing at 95 ℃ for 120 s; 35 cycles of 95 ℃ for 60 s, 50 ℃ for 60 s, and 72 ℃ for 60 s; a final elongation step at 72 ℃ for 300 s; and refrigeration at 4 ℃ until removal and storage.

In the second step, Illumina adaptors, barcodes, and indices were added to the amplicons. The same PCR reaction mix was used with custom primers targeting the primer extension in the first PCR. The thermal cycling





conditions were as follows: initial denaturing at 95 ºC for 180 s; 8 cycles of 95 ºC for 30 s, 55 ºC for 30 s, and 72
       ºC for 30 s; a final elongation step at 72 ºC for 300 s; and refrigeration at 4ºC until removal and storage. Amplicons
       were cleaned with 0.7x AMPure XP magnetic beads (Beckman-Coulter, Brea, CA, USA), pooled, and quantified
       before being sent to the UC Davis DNA Technologies Core Facility (Davis, CA, USA) for Illumina MiSeq 2x250
       bp (16S rRNA) or 2x300 bp (*mcrA*) sequencing. Ten 16S rRNA and four *mcrA* samples were randomly chosen
for duplicate amplification. The average weighted UniFrac distance between duplicate 16S rRNA samples was
       0.067. Negative (molecular grade water) and positive (mock communities of known composition) controls were
       processed and sequenced in parallel with the samples. Lack of DNA contamination in the RNA extracts was
       confirmed by processing RNA extracts (without reverse transcription) in parallel and seeing no visible
       amplification on a gel after the second PCR.

**2.6 Phylogenetic analysis of 16S rRNA and *mcrA* genes and transcripts**

       Demultiplexed sequences were trimmed with cutadapt (v. 2.10; Martin, 2011), then filtered and
       processed using the R (v. 4.2.1) package DADA2 (v. 1.26.0; Callahan et al., 2016). Reads were trimmed to 216
       (16S rRNA) or 260 and 230 (*mcrA*; forward and reverse reads, respectively) base pairs, with those containing
       more than 2 expected sequencing errors removed. Amplicon sequence variants (ASVs) were then inferred from
filtered reads. Phylogenetic classification of 16S rRNA ASVs was based on the SILVA SSU database (v. 132;
       Quast et al., 2013). Classification of *mcrA* ASVs was determined manually based on placement on a reference
       tree (detailed below). On average, 937 16S rRNA reads were recovered per blank sample, while 20,853 16S rRNA
       reads were recovered per *in-situ* sample. On average, 4 *mcrA* reads were recovered per blank sample, while 53,021
       *mcrA* reads were recovered per *in-situ* sample.

**2.7 Classification of *mcrA* ASVs**

       To classify *mcrA* sequences, the tool EPA-ng (v 0.3.8) was used to place the ASVs onto a reference tree.
       Reference tree sequences included published sequences from cultured methanogens or ANME, as well as 6 of the
       ASVs themselves (those that represented >10% of any sample's total sequences and had no cultured match above
       90% similarity in NCBI and would therefore be represented poorly on the reference tree). Reference sequences
were aligned with MAFFT (v. 7.490) and incorporated into a RAxML (v. 8.2.12) best-scoring ML reference tree
       with 100 bootstraps. After ASV placement with EPA-ng, relative abundances of those ASVs in each sample were
       displayed in a heatmap using ggtree (v. 3.6.2). Relative abundances of ASVs placed on internal tree nodes were
       divided among tip nodes associated with that internal node.





### 2.8 Sequence analysis and statistical methods

Non-metric multidimensional scaling (NMDS) of microbial communities was carried out based on the weighted UniFrac distance metric (Lozupone, 2007) using the R package "vegan" (v. 2.5-7; Oksanen et al., 2020). Analysis of similarity (ANOSIM) was used to determine the significance of microbial community differences between groups of samples, also based on the weighted UniFrac distance metric. The R package "DESeq2" (v. 1.38.3; Love et al., 2014) was used to test if ASVs were significantly enriched in abundance with time (0 months

vs. 6 months) across Clam Field seep incubations.

### 2.9 Droplet digital PCR

Droplet digital PCR (ddPCR) was used to quantify abundances of *mcrA* genes and transcripts in Seep and Background-5m cores from Clam Field and Extrovert Cliff using the primer pair mcrA_F/mcrA_R (Luton et al., 2002; Dekas et al., 2016). For comparison, *mcrA* genes and transcripts were also quantified in a seep and a

background core from an alternative seep site (New England seep) on the northern U.S. Atlantic Margin. 25 µL PCR reactions were performed containing 1 µL of forward and 1 µL of reverse primers (5 µM concentration), 12.5 µL EvaGreen SuperMix (2x, Bio-Rad Laboratories, Hercules, CA, USA), 9 µL DNase-free water, 0.5 µL bovine serum albumin (2.5 µg/µL), and 1 µL diluted DNA or cDNA template. Droplets were generated on a QX200 Droplet Generator (Bio-Rad) at the Stanford Functional Genomics Facility (Stanford, CA, USA) using

droplet generation oil for EvaGreen (Bio-Rad). Thermal cycling was performed immediately afterward on a C1000 Touch Thermal Cycler, with thermal cycling conditions as follows: initial denaturing at 95 ºC for 300 s; 45 cycles of 95 ºC for 60 s, 52 ºC for 90 s, and 72 ºC for 75 s; signal stabilization steps at 4 ºC for 300 s and 90 ºC for 300 s; and a final 10 ºC hold overnight. The overall ramp rate was set at 1 ºC/s.

Droplets were read with the QX200 Droplet Reader (Bio-Rad). Threshold fluorescence values were

initially inspected using QuantaSoft software (Bio-Rad), but the values were later adjusted using the minimum density clustering method, which better separated droplet clusters upon manual inspection. Amplicon copy numbers per well were then converted to copies/g dry sediment. Technical replicates were run for a randomly selected half of the samples to confirm the precision of our assay. Results of the replicate runs are shown in Supplementary Figure S1.



### 2.10 DAPI staining and microscopy

To visualize putative ANME aggregates under the microscope, cells were fixed aboard in 4% paraformaldehyde according to the protocol of Dekas et al. (2009). Fixed sediment was diluted 1:20 in phosphate buffered saline (PBS), sonicated 3 x 10 s at an amplitude of 30 on a Q500 sonicator (Qsonica, Newtown, CT, USA), and floated on top of a preestablished Percoll-PBS gradient (protocol described in Orphan et al. (2002) and

Dekas and Orphan (2011). In total, 1 mL sonicated sample was added to 9 mL Percoll-PBS. After floating the sample, the mixture was centrifuged at 4780 rpm for 15 min at 4 ºC, the supernatant filtered through a 25 mm 3 µm polycarbonate filter (Millipore, #TSTP02500) backed with a glass microfiber filter (Cytiva Whatman, GF/F, #1825-025) under low (<5 psi) vacuum, and washed with 1 mL PBS, then 1 mL 100% EtOH. Each filter was sectioned with a razor blade and stained with 4′,6-diamidino-2-phenylindole (DAPI; #D9542, Sigma-Aldrich,

Darmstadt, Germany) before visualization on an inverted fluorescence microscope (Nikon Eclipse Ti). The number of aggregates in 100 random fields of view at 400x magnification were counted, corresponding to a detection limit (<1 aggregate per 100 fields of view) of 8.58 x $10^4$ aggregates ($g^{-1}$ dry sediment).

### 2.11 Methane diffusive flux calculations

The amount of methane potentially diffusing out of sediments in the absence of biological consumption

was estimated according to Fick's laws of diffusion (Boudreau, 1997). Equation 1 calculates the diffusion rate (J) in mmol $m^{-2}$ $yr^{-1}$:

$$J = -\varphi D_s \frac{dC}{dz} \qquad (1)$$

where φ is the sediment porosity, $D_s$ is the sediment diffusion coefficient (in $m^2$ $yr^{-1}$), and dC/dz is the methane concentration gradient over sediment depth (in mmol $m^{-3}$ $m^{-1}$). The sediment diffusion coefficient, $D_s$, can be

calculated via Equation 2:

$$D_s = \frac{D_0}{1 + n(1 - \varphi)} \qquad (2)$$

where $D_0$ = 1.4 x $10^5$ $cm^2$/s (the initial diffusion coefficient of methane at 20 ºC; Boudreau, 1997), n = 3 (the lithology factor of silty clay), and φ is the sediment porosity.





**3 Results**

**3.1 Geochemical environment**

Both cold seep sites contained moderate to high concentrations of porewater methane in all cores collected within the putative seep boundaries (i.e. both the Seep and Seep-Edge cores; Fig. 2b). Methane concentrations were two orders of magnitude higher at Extrovert Cliff (ranging from ~3000 to ~6400 µM) than at Clam Field (ranging from 25 µM to 183 µM), but even Clam Field contained methane concentrations roughly 4

times higher than the surrounding background sediment. At Clam Field, methane concentrations increased with depth in the sediment; methane concentrations at Extrovert Cliff, in contrast, showed no trend with depth, but were consistently high throughout the sediment core. At both sites, methane concentrations in background sediments were measurable, and ranged from 21 to 56 µM.

Sulfide concentrations at both cold seep sites were also elevated within the seep boundaries (Fig. 2b). At

Extrovert Cliff, sulfide concentrations were elevated at all sediment depths (ranging from 2.7 to 12.7 mM), with no depth trend. At Clam Field, sulfide concentrations were below detection at the surface, and increased with sediment depth (up to 11.2 mM).

With sediment depth, $\delta^{13}$C-DIC decreased at Clam Field (Fig. 2b) from -3.1 to -17.7‰, with the lowest values found in the deepest depths (Fig. 2). At Extrovert Cliff, $\delta^{13}$C-DIC was consistent with sediment depth (from

-9.8 to -11‰), though at both sites, $\delta^{13}$C-DIC was more negative in cores collected within the seep than in cores collected in background sediments (from -0.1 to -4.9‰).

**3.2 Quantification of *mcrA* genes and transcripts**

To evaluate methane cycling both on and off each cold seep, and to compare those results with methane and sulfide concentrations, we quantified *mcrA* genes and transcripts with droplet digital PCR (ddPCR). *mcrA*

encodes the alpha subunit of methyl coenzyme-M reductase and is a marker gene for both methanogens and anaerobic methanotrophs (Luton et al., 2002; Hallam et al., 2003; Krüger et al., 2003); therefore, quantification of the gene cannot distinguish between the two metabolisms. *mcrA* gene copies (g⁻¹ dry sediment) were highest in Extrovert Cliff seep sediments and peaked between 2.5-5 cmbsf at $5.3 \times 10^7$ copies (g⁻¹ dry sediment) – roughly the same sediment depth where methane concentrations began to decrease up core (Fig. 3). Gene copies of *mcrA*

were an order of magnitude less abundant at Clam Field seep – peaking in the uppermost sediment horizons at concentrations of $4.2 \times 10^6$ copies (g⁻¹ dry sediment). However, gene copy numbers at Clam Field seep were still above the average of $7.4 \times 10^5$ *mcrA* gene copies (g⁻¹ dry sediment) in background sediments from either site.





Transcript copies of *mcrA* (per g dry sediment) were elevated in Extrovert Cliff seep sediments compared to background sediments (by a factor of 2) but were equivalent between Clam Field seep and background sediments.


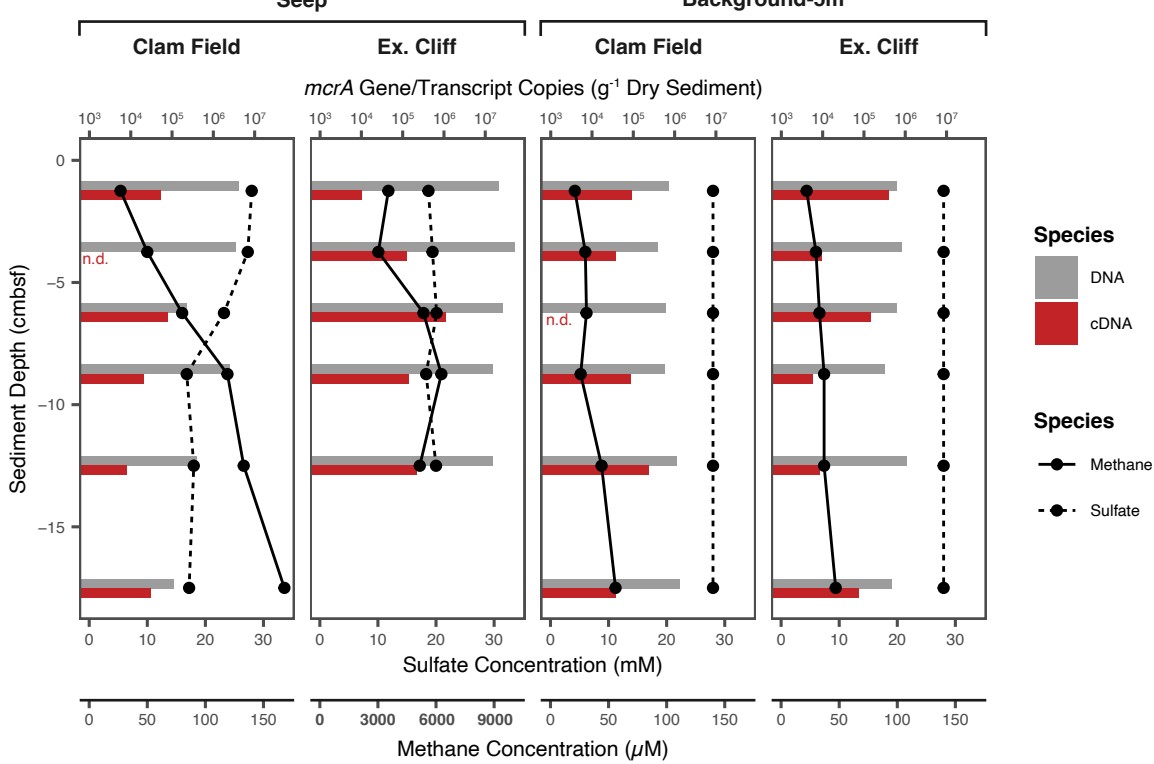

**Fig. 3: Methane and calculated sulfate (28mM - sulfide) concentration (lines), and *mcrA* gene (DNA) and transcript (cDNA) concentration (bars) with sediment depth in Seep and Background-5m cores from both study sites. (*mcrA* gene and transcript concentrations were not measured in Seep-Edge nor Background-**

**100m cores.) Note the difference in x-axes values for methane concentration in the Extrovert Cliff Seep panel (axes labels in bold).**

n.d. – measured, but not detected in a given sample



### 3.3 Community composition at Monterey Bay cold seeps via 16S rRNA amplicon sequencing

To characterize the microbial communities at these sites, we performed 16S rRNA amplicon sequencing in the seep (Seep and Seep-Edge) and background samples (5 m and 100 m from seepage) from both sites. In the 21 seep samples, we recovered 6,402 unique 16S rRNA ASVs. The most abundant of these were members of the phyla Bacteroidota, Campylobacteria, Desulfobacterota, Chloroflexi, Proteobacteria (mainly Gammaproteobacteria), and Verrucomicrobiota. The relative abundances of Campylobacteria and Chloroflexi

tended to increase with sediment depth in a core, while the relative abundance of Bacteroidota decreased with sediment depth (Fig. 4a). Across 24 background samples, we recovered 12,731 unique 16S rRNA ASVs. Microbial communities in the background sediments from both sites were similar to one another at the phylum level (ANOSIM: p-value = 0.066) with Bacteriodota, Chloroflexi, Desulfobacteria, and Proteobacteria as the most abundant groups (Fig. 4a). There was no significant difference between microbial communities in Background-

5m (5 m from seepage) and Background-100m cores (100 m from seepage) – neither at the phylum level (ANOSIM: p-value = 0.102), nor at the ASV level (ANOSIM: p-value = 0.084). Consistent with the typical "seep microbiome" (Ruff et al., 2015; Semler et al., 2022), Caldatribacteriota, Campylobacterota, and Verrucomicrobiota were enriched in seep cores in comparison to background cores (>3 times greater) at both sites, while Desulfobacterota and Proteobacteria (mainly Gammaproteobacteria) had particularly high relative

abundances in background cores in comparison to seep cores (roughly 2 times greater; Fig. 4a). While relative potential activity (as inferred by 16S rRNA (RNA)) mirrored many of the trends seen in the DNA, reads from several phyla, including Halobacterota and Desulfobacterota, were more than twice as abundant in the RNA fraction of the seep core at Extrovert Cliff; Halobacterota was 40 times more abundant in the RNA analysis of the seep core at Extrovert Cliff.

Halobacterota – the phylum containing the majority of methanogenic and anaerobic methanotrophic taxa (including ANME archaea) – were present in very low relative abundances in the 16S RNA gene dataset. At Clam Field, these organisms were not detected in seep or background sediments, with the exception of extremely low (<0.1%) abundances of ANME-3 in the two upper sediment horizons of the Seep-Edge core (Fig. 4b). No Halobacterota reads were detected in cDNA at Clam Field. At Extrovert Cliff, all seep sediment horizons

contained low levels of Halobacterota (<0.3% of the community), with the majority assigned to ANME-2c or ANME-3. A small number of methanogenic Methanomicrobia were also present in the lowest sediment depths of the Extrovert Cliff Seep core. While Halobacterota were not highly abundant at Extrovert Cliff, their relative potential activity was high—up to ~18% of total community cDNA in some cores. This transcriptional activity was almost entirely associated with ANME-2c and was highest in the 5-7.5 cmbsf depth horizons, consistent with









**Fig. 4 (previous page): Relative abundance (%) of Archaea and Bacteria phyla (A) and of methane-cycling and sulfate-reducing subgroups within Halobacterota and Desulfobacterota, respectively (B), across sampled sediment horizons from Seep, Seep-Edge, Background-5m, and Background-100m cores, as inferred by 16S rRNA gene (DNA) and 16S rRNA (cDNA) sequencing. Boxes surround each individual**
**core, with sediment depth in each core increasing from left to right. Note the variable relative abundance percentages on the y axis.**

the location of the peak in *mcrA* transcripts in the ddPCR data. Relative abundance of rRNA is a not precise proxy for relative activity between taxa, but it can indicate which taxa are likely translationally active (Blazewicz et al.,
2013). We refer to the detection of rRNA as reflecting 'potential' activity to emphasize the limitations of this proxy.

Desulfobacterota – the phylum comprising known ANME symbionts as well as other free-living sulfate reducers common at cold seeps – comprised roughly 20% of the Clam Field microbial community and roughly 10% of the Extrovert Cliff microbial community (Fig. 4b). The most common subgroups within the
Desulfobacterota were Desulfobacteria (including Seep-SRB1), Seep-SRB2 (at Extrovert Cliff only), and Seep-SRB4 (particularly at Clam Field). Seep-SRB1, the group containing many known obligate ANME symbionts, were found in low abundances at both seep sites, ranging from 0.70-2.04% of sequences in Clam Field seep cores, and from 0.73-1.26% of sequences at Extrovert Cliff. Also present – particularly at Clam Field – were Desulfomonadia, a group associated with sulfur- and iron-reducing capabilities (Ravenschlag et al., 1999; Wunder
et al., 2021). At Clam Field, the cDNA profiles of Desulfobacterota generally mirrored the DNA profiles, with Desulfobacteria dominating the reads overall, and Desulfobulbia, Seep-SRB4 and Seep-SRB1 showing relative potential activity peaks in the top, mid, and bottom portions of the cores, respectively. At Extrovert Cliff, Desulfobacterota comprised roughly 3 times more of the cDNA reads than they did for the DNA, mirroring the high relative potential activity observed for the Halobacteria there. The increase was largely due to Seep-SRB2,
which alone comprised up to 30% of the cDNA reads in the 5-7.5 cmbsf depth horizon in both cores. This is consistent with the previously described association of members of ANME-2c and Seep-SRB2 (Krukenberg et al., 2018; Kleindienst et al., 2012) and suggests active anaerobic methanotrophy by these groups at this depth.

**3.4 Community composition at Monterey Bay cold seeps via *mcrA* amplicon sequencing**

To provide greater insight into the diversity of methane-cycling microorganisms at these sites, and
specifically increase our ability to detect low-abundance methanotrophs, if present, we sequenced *mcrA* genes at




both sites. Sequencing the *mcrA* gene specifically increased our detection limit of methane-cycling organisms by nearly 3 orders of magnitude; as we recovered on average 39,044 reads per sample using *mcrA* sequencing versus 61 reads of putative methane-cycling organisms per sample with 16S rRNA gene sequencing. Within Extrovert Cliff seep samples, the *mcrA* gene results were generally consistent with the 16S rRNA results, with *mcrA*

sequences affiliated with ANME-3 detected in shallow depth horizons and ANME-2c and methanogenic archaea appearing deeper in the cores. Interestingly, within the two seep cores at Extrovert Cliff (both Seep and Seep-Edge), a single ASV affiliated with the ANME-2c group comprised ~50% of the total *mcrA* reads, peaking between 5 and 10 cmbsf (Fig. 5). Similar observations of local dominance of individual ANME ASVs have been made at other seep sites previously (Semler et al., 2022).

The *mcrA* sequences from Clam Field revealed a more nuanced perspective of methane cycling than the 16S rRNA sequences, with seep samples containing ANME-1 as well as a variety of putatively methanogenic archaea. Notably, most of these, including the ANME-1 reads, were more relatively abundant in the background samples at Clam Field than the seep samples, indicating they were not enriched by increasing methane concentrations. The one exception was ASV.2, which was the most abundant ASV at Clam Field seeps (alone

comprising roughly 45% of seep ASVs) and was almost absent from background sediments (0.1% of background ASVs). ASV.2 has no cultured relative above 90% similarity in NCBI, but clusters with sequences from *Methanohalophilus halophilus* and *Methanomethylovorans hollandica*, anaerobes involved in methylotrophic methanogenesis (Lomans et al., 1999). The most highly abundant ASV (ASV.3) in Clam Field and Extrovert Cliff background sediments is most closely related to strain MO-MCD, also a methylotrophic methanogen, belonging

to the genus *Methanococcoides* (Singh et al., 2005). Although this organism is also present within the Clam Field seep, the clear difference in distribution between ASV.2 and ASV.3 across the seep boundary highlights niche separation between putatively similar methylotrophic methanogens at these sites. In general, while ANME affiliated sequences dominated the *mcrA* dataset from seep samples at Extrovert Cliff (85.6% of *mcrA* reads), reads associated with methylotrophic methanogens dominated both Clam Field (>80% of *mcrA* reads; only 5.6%

were affiliated with ANME) and the background sediments from each site (>75% and >60% of reads from background cores at Extrovert Cliff and Clam Field, respectively). We also attempted to sequence *mcrA* transcripts at Clam Field (Table S4), but *mcrA* expression was below the limit of detection (no visible amplification of the cDNA on a gel).

By multiplying the *mcrA* copy number in samples (as determined by ddPCR) by the proportion of
ANME-affiliated *mcrA* reads in that sample (as determined by the amplicon sequencing analysis using the same primer set), ANME cell numbers in each sample were estimated (Table S5). In total, ANME were not enriched in







**Fig. 5 (previous page): Relative abundance (%) of all ASVs inferred from mcrA gene sequencing (DNA) at Clam Field and Extrovert Cliff, and their distribution on an mcrA reference tree. Sequences on the tree**

**were aligned with MAFFT (v. 7.490) and incorporated into a RAxML (v. 8.2.12) best-scoring ML reference tree with 100 bootstraps. Heatmap values were calculated by adding the relative abundance of all ASVs assigned to that branch by EPA-ng (v. 0.3.8). Tree was rooted with Bathyarchaeota sp. (KT387810). The scale bar indicates the average number of amino acid substitutions per site, and the filled circles signal nodes with at least 60% (grey) or 80% (black) bootstrap support.**


Clam Field seep sediments ($5.0 \times 10^4$ to $8.2 \times 10^6$ ANME cells g$^{-1}$ dry sediment) relative to background sediments ($8.3 \times 10^5$ to $6.9 \times 10^7$ ANME cells g$^{-1}$ dry sediment), and ANME *mcrA* genes were attributed primarily to ANME-1 (Fig. 5). ANME cell numbers were estimated to be roughly 4 orders of magnitude higher at Extrovert Cliff seep ($8.2 \times 10^8$ to $4.7 \times 10^9$ ANME cells g$^{-1}$ dry sediment) than at Clam Field seep (Table S5), with ANME *mcrA* genes

attributed primarily to ANME-2c and ANME-3 (Fig. 5).

### 3.5 Visualization of ANME aggregates

To visualize and quantify potential aggregates of ANME archaea and sulfate reducing bacteria from these seeps, we examined DAPI-stained cells from methane and sulfate replete sediments at Clam Field (7.5-10 cmbsf) and Extrovert Cliff (5-7.5 cmbsf) seeps. While not a taxa-specific assay, visualizing aggregates, or their absence,

provides independent support for the presence or absence of aggregate-forming ANME-2 and ANME-3 archaea in the molecular data. In sediments from Extrovert Cliff, where ANME-2 and -3 were found in the DNA (0.2% of 16S rRNA gene reads) and RNA analyses (10.1% of 16S rRNA reads), we detected putative ANME cell aggregates at a concentration of $4.35 \times 10^6$ aggregates (g$^{-1}$ dry sediment) (Supplementary Fig. S2). Considering ANME aggregates can contain anywhere from tens to thousands of cells, this value is roughly consistent with the

estimate of ANME density derived from the molecular data. Cell aggregates were not found at Clam Field, consistent with the lack of ANME-2 and dearth of ANME-3 sequences detected at this site. Our aggregate detection limit corresponded to $8.58 \times 10^4$ aggregates (g$^{-1}$ dry sediment).

### 3.6 Changes in microbial community and geochemistry with methane addition

To investigate whether sulfate-coupled methane oxidation was occurring at Clam Field and determine

whether canonical ANME archaea could be enriched there, Clam Field sediments were incubated under varying concentrations of methane. Sulfide concentrations increased continuously over six months in sediment from the



seep center, up to 10 mM, indicating active sulfate reduction (Supplementary Fig. S4). A smaller increase (~1-2 mM) was observed in sediments from the "Seep-Edge" core. However, there was not a statistically significant difference in sulfide production with or without methane in either core, indicating that sulfate reduction was not

methane-dependent. We therefore did not see evidence of sulfate-coupled methane oxidation in these sediments, in contrast to previous observations at other seeps (e.g., Hydrate Ridge (Nauhaus et al., 2002, 2007), Eel River Basin (Dekas et al., 2009), and the Costa Rica Margin (Dekas et al., 2014)).

Consistent with a lack of methane oxidation, ANME archaea were not enriched during the 6 month incubation, as assessed by 16S rRNA sequencing (DNA and RNA; Supplementary Fig. S3). Taxa that were

significantly enriched after six months with methane included several ASVs within the Desulfobacterota, specifically Desulfovibrio, Desulfobulbus, Desulfuromonas, and Seep-SRB4 (Supplementary Fig. S5A). A putatively methylotrophic methanogen in the genus *Methanococcoides* was also significantly enriched relative to the pre-incubation community. Aerobic sulfide-oxidizing groups, including Thiobeggiatoa, Colwellia, and Thiomargarita, significantly decreased in abundance over 6 months with methane. Without methane, none of the

above taxa were significantly enriched or unenriched after 6 months of incubation time (Supplementary Fig. S5B). A single Verrucomicrobia ASV had significantly increased in abundance and a single Proteobacteria ASV (from the family Beggiatoaceae) had significantly decreased in abundance.

**3.7 Comparison to Previously Characterized Sites**

To directly compare Monterey Bay seep microbial communities with canonical seep communities, we compared them to those of four seeps along the U.S. Atlantic Margin (USAM). First, we compared the 16S rRNA gene profiles of the Monterey Bay seeps to that of the USAM seeps using non-metric multidimensional scaling (NMDS). Monterey Bay seep samples formed a separate cluster from the Atlantic samples (Fig. 6a), and communities from Extrovert Cliff were nested within those from Clam Field. While the primary axis of the NMDS

plot was defined by sediment depth (Fig. 6b), the secondary axis was defined by geographic region. When including background samples from all sites, background samples from Monterey Bay clustered together within background sediment communities sequenced from the USAM (Fig. 6c). As in the seep-only community comparison, the primary axis of the plot was defined by sediment depth (Fig. 6d), though the secondary axis was instead defined by environment type (seep vs. background), rather than sampling region.





We also measured *mcrA* concentrations via ddPCR at New England seep – one of the four USAM seeps
        – to better contextualize the trends in *mcrA* abundance observed within Monterey Bay. At New England seep,

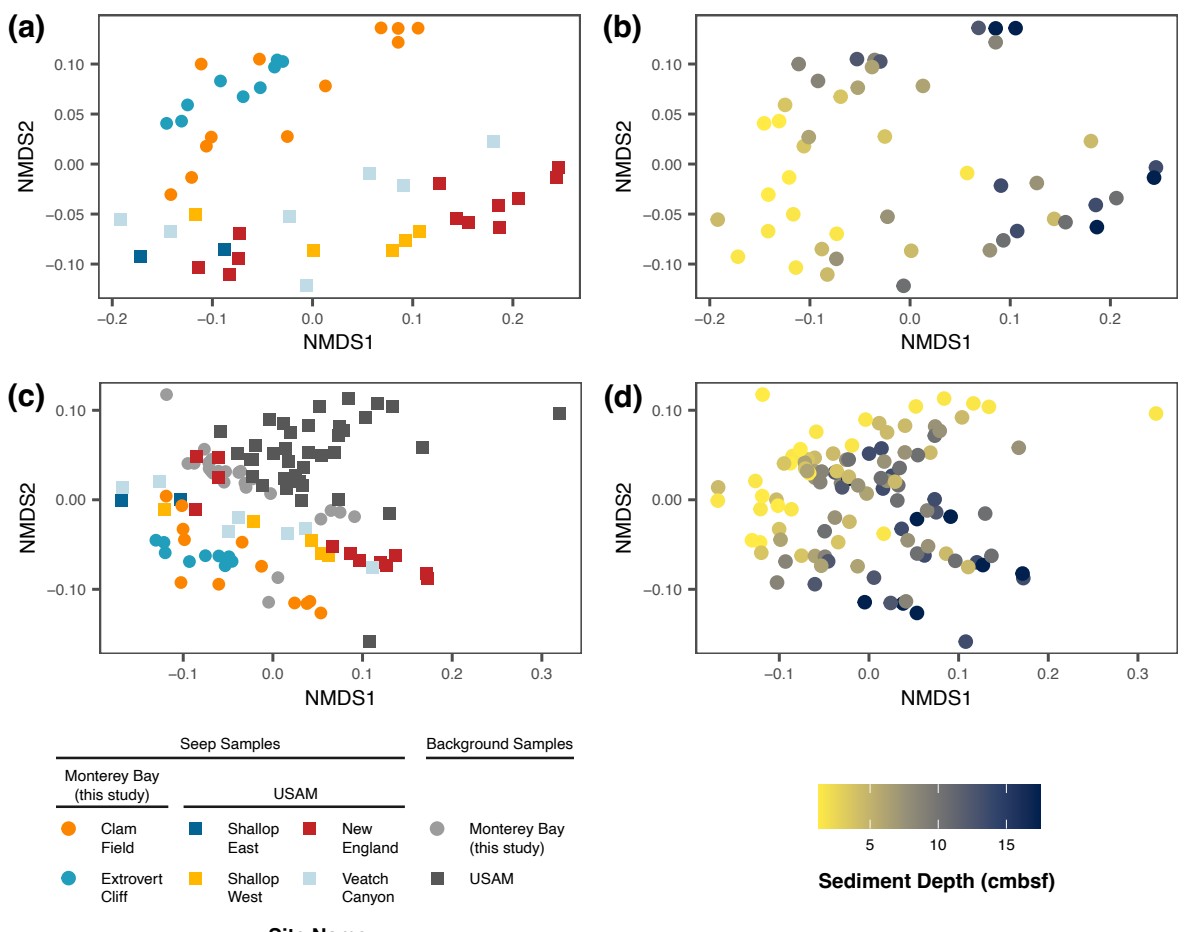

**Fig. 6: Non-metric multi-dimensional scaling (NMDS) of seep samples only (a-b) and of both seep and background samples (c-d) from Monterey Bay (Clam Field and Extrovert Cliff; this study), as well as from**
**U.S. Atlantic Margin seep and background samples (Semler et al., 2022). Samples colored by site (a, c) and by sediment depth (b, d). NMDS was based on a weighted UniFrac distance metric and was inferred by 16S rRNA gene sequencing.**

        where ANME archaea had represented 13.4% of the microbial community in 16S rRNA gene surveys on average
(Semler et al., 2022), we found that *mcrA* gene copy numbers reached 2.5 x 10$^8$ copies (g$^{-1}$ dry sediment) at the





deepest sediment depths, outnumbering those at both Extrovert Cliff and Clam Field by one and two orders of magnitude, respectively (Supplementary Fig. S1). However, *mcrA* transcript copy numbers at New England seep were comparable to those at Extrovert Cliff, though peaked at lower sediment depths (15+ cmbsf).

We also chose two USAM seeps (New England and Shallop Canyon East) at which to visualize putative ANME aggregates. Again, fixed sediments from methane and sulfate replete depths (0-3 cmbsf) at these two sites were examined under an epifluorescence microscope with a DAPI stain. In the chosen horizons, ANME had comprised 4.8% (New England) and 0.5% (Shallop Canyon East) of the microbial community in 16S rRNA gene surveys (Semler et al., 2022). Consistent with their higher relative abundances in the molecular data compared to the Monterey seeps, we discovered higher concentrations of putative aggregates in these sediments by

microscopy: $1.15 \times 10^7$ and $1.34 \times 10^6$ aggregates ($g^{-1}$ dry sediment), respectively.

**3.8 Methane diffusive flux**

The porewater methane concentrations at Clam Field decreased approximately linearly up core, indicative of diffusive flow between the methane-rich sediments and the methane-poor overlying water column. This is in contrast to a concave up trend, which is indicative of biological methane consumption and therefore

AOM (Reeburgh 1976; Martens and Berner, 1977; Ward et al., 1987). The lack of methane-dependent sulfide production in sediments from this site, as well as the scarcity of ANME archaea, further support a lack of significant biological methane consumption. Although we cannot exclude the possibility of low levels of biological oxidation, if we assume diffusion is the only mechanism removing methane from seep sediment, we can calculate the diffusive flux to determine the potential amount of methane released to the water column at this

site. The methane concentration gradient over sediment depth (dC/dz) was determined from the average slope of the linear relationship between methane concentration and sediment depth (Supplementary Fig. S6). Calculated fluxes were 17.8 and 18.5 mmol methane $m^{-2}$ $yr^{-1}$ for the Clam Field "Seep" and "Seep-Edge" cores, respectively (Table S6).

**4. Discussion**

The scarcity of 16S rRNA and *mcrA* sequences belonging to ANME archaea in the Monterey Bay seeps, and particularly at Clam Field, is unexpected and intriguing. In typical methane seep sediments, a characteristic suite of microbial community members – termed the "seep microbiome" – is present and remarkably consistent despite geographical separation (Ruff et al., 2015; Semler et al., 2022). This community consists most notably of



ANME archaea (including ANME-1a, -ab; ANME-2a, -2b, -2c, -2d; and ANME-3) and SRB (including members
of the Seep-SRB1 and Seep-SRB2 in the Desulfobacterales, *Desulfobulbus* and Seep-SRB4 in the
Desulfobulbales; and thermophilic HotSeep-1). Other commonly occurring microbial groups at seeps include
sulfide-oxidizing and aerobic methane-oxidizing Gammaproteobacteria, and the putatively methanotrophic JS1
lineage of Atribacterota. While Monterey Bay is a somewhat geographically isolated environment, seep
communities are thought to assemble primarily deterministically – based on local geochemical variables like
methane, sulfate/sulfide, and ammonium concentrations, and do not appear to be limited by dispersal (Semler et
al., 2022). It is therefore unlikely that the low levels of ANME archaea in Monterey Bay is a result of dispersal
limitation.

At Extrovert Cliff, ANME-SRB consortia members are not highly abundant (<0.3% ANME in any given
sample), but they have high potential relative activity and active methane oxidation is likely. ANME archaea
typically comprise a large portion of seep communities, for instance, 13.4% at New England Seep on the USAM.
Methane concentrations are extremely high at Extrovert Cliff, and most geochemical variables show no obvious
depth-related trend in the top 15 cm (Fig. 2b). Other studies have noted the high, though temporally variable, fluid
flux here (LaBonte et al., 2007; Füri et al., 2009), which potentially homogenizes the upper sediment layers and
masks depth-related trends. *mcrA* gene copy numbers, which reached a maximum of 5.3 x $10^7$ copies (g$^{-1}$ dry
sediment), were moderately low as compared to concentrations at other seeps, for instance, $10^9$ *mcrA* gene copies
(g$^{-1}$ wet sediment) in a seep in the Nankai Trough  (Nunoura et al., 2006), $10^8$ copies (g$^{-1}$ wet sediment) in a seep
in the Kumano Knoll (Miyazaki et al., 2009), $10^7$ copies (g$^{-1}$ wet sediment) in a seep core from the South China
Sea (Niu et al., 2017), and 2.5 x $10^8$ copies (g$^{-1}$ dry sediment) measured in this study for USAM seep sediment
(Fig. S1). However, ANME-2c comprises >10% of 16S rRNA sequences (RNA fraction) in some Extrovert Cliff
seep samples, coincident with likely syntrophic Seep-SRB2 comprising >20% (Fig. 4b). At Extrovert Cliff, it is
therefore likely that a small number of extremely active ANME-SRB consortia (particularly ANME-2c and Seep-
SRB2) perform substantial methane oxidation.

At Clam Field, ANME were both low in abundance and low in potential relative activity, as ANME
sequences were not detected in the RNA from this site at all. While extraction, sequencing, and/or primer biases
can cause artificial underestimates, our successful recovery of ANME 16S rRNA and *mcrA* genes/transcripts at
Extrovert Cliff here and previously at U.S. Atlantic Margin sites using the same protocols (Semler et al., 2022)
suggest a technical artifact is unlikely. Furthermore, our use of both 16S rRNA and *mcrA* primer sets, as well as
microscopy to detect ANME-typical morphologies, reduce the possibility that a diverged ANME lineage was
overlooked by a particular primer set. Though 5.6% of *mcrA* genes at Clam Field were affiliated with ANME-1,



*mcrA* gene concentrations reached a maximum of 4.2 x $10^6$ copies ($g^{-1}$ dry sediment) throughout the seep, indicating that the overall abundance of ANME-1 was low. Additionally, the relative and inferred absolute abundance of the ANME-1 ASVs found within Clam Field seeps were actually higher in background cores, consistent with the possibility that this group is not exclusively methanotrophic (Lloyd et al., 2011; Kevorkian et al., 2021). While it is possible that biological methane oxidation is being performed by organisms not identified

as methane oxidizers at this site—with *mcrA* ASV.2 the most likely candidate—the lack of evidence for active methane oxidation in the porewater methane profile and the sediment incubations over time makes this possibility unlikely.

A lack of ANME archaea is unexpected in methane-rich marine sediments. Methane concentrations at Clam Field are elevated far above background concentrations (up to 183 µM), and although they are lower than

at Extrovert Cliff, they are higher than at 3 of 4 USAM sites where ANME are abundant (Semler et al., 2022). A dearth of ANME sequences in methanic sediments has been observed at just a few other methane-rich sites (Goffredi et al., 2008; Ruff et al., 2019; Thurber et al., 2020), but these observations have been made only in recently perturbed environments such as whale falls or young, newly emerged (< 1 year) seeps. Clam Field, in contrast, has a documented history of elevated methane concentrations and of characteristic seep macrofauna

going back nearly three decades (Barry et al., 1996, 1997; Lorenson et al., 2002). Despite 1) the lack of ANME dispersal limitation globally (Ruff et al., 2015; Semler et al., 2022), 2) the success of these protocols at detecting ANME at other seeps, and 3) the current and historically high concentrations of methane and sulfide at this site, we do not detect an active ANME population at Clam Field. Therefore, an external factor likely limits the presence of ANME.

The location of Clam Field seep may be the key to its unique microbial assemblage, as historical data indicates that this site experiences non-methane hydrocarbon inputs in addition to methane. The site is situated within the Monterey Bay Fault Zone and, in particular, within sediments where the hydrocarbon-rich Monterey Formation crops out. Lorenson et al. 2002 recovered two oil-stained rocks at this site, roughly 20 cmbsf, which were later measured to contain n-chain alkanes from $C_{16}$-$C_{35}$, along with a complex mixture of unresolved

hydrocarbons. Orange et al. (1999) and Stakes et al. (1999) described an authigenic carbonate sample with a distinct aromatic hydrocarbon odor. Furthermore, methane here has a distinct thermogenic fingerprint; reported $^{13}C$-$CH_4$ values were heavier than at other measured cold seep sites – -50 to -55‰ rather than -70 to -85‰ (Orange et al., 1999; Lorenson et al., 2002).

The $\delta^{13}C$-DIC concentrations we measured at Clam Field (-3 to -18‰) are consistent with the oxidation

of non-methane hydrocarbons. The values are heavier than at a typical cold seep with active methanotrophy



[typically -40 to -30‰ (Paull et al., 2000; Liu et al., 2020; Sauer et al., 2021), even with a thermogenic methane source (Sauer et al., 2021)], though lighter than those in background sediments at this site (0.9 to -4.9‰; Fig. 2b). The oxidation of non-methane hydrocarbons, which have heavier isotopic signatures (-33 to -29‰ for oil) than methane (Joye, 2020), leads to DIC that is less $^{13}$C depleted, as is observed at Clam Field. However, at Extrovert

Cliff, the $\delta^{13}$C-DIC is also only moderately depleted (-9.8 to -11‰), despite signs of active methanotrophs. It is possible that the oxidation of hydrocarbons is common at both sites and dominates the $\delta^{13}$C-DIC signal despite the co-occurrence of methane oxidation at Extrovert Cliff. But, a moderate $\delta^{13}$C-DIC signal derived from isotopically light methane-derived DIC mixed with seawater-DIC (consistent with turbulence at a higher flux seep system) could also result in the observed intermediate values at Extrovert Cliff.

550       The presence of oil and other non-methane hydrocarbons has been previously documented to play a role in shaping microbial communities at seeps (Orcutt et al., 2010; Vigneron et al., 2017), and some of the lineages responsible for their oxidation are abundant in our dataset. In Gulf of Mexico seep sediments, seeps with input of non-methane hydrocarbons were typically characterized by specific Desulfobacterota lineages (Vigneron et al., 2017), which can be involved in the anaerobic oxidation of non-methane hydrocarbons with sulfate as an electron

acceptor (Kleindienst et al., 2014; Vigneron et al., 2017; Joye, 2020). The *Desulfococcus/Desulfosarcina* (DSS) clade comprises organisms that degrade short-chain alkanes, as well as degraders of mid-chain or long-chain alkanes, alkenes, or aromatic compounds (Aeckersberg et al., 1998; Harms et al., 1999; So and Young, 1999; Meckenstock et al., 2002). Hydrocarbon degradation genes, including *assA* and *bssA*, are also found in members of the non-DSS *Desulfobacteraceae*, *Syntrophobacteraceae*, and *Desulfatiglans* (Widdel and Grundmann, 2010;

Vigneron et al., 2023), and members of *Desufatiglans*, Seep-SRB1d, and Seep-SRB4 have been continually overrepresented in sediments affected by seepage of non-methane hydrocarbons (Kleindienst et al., 2014; Vigneron et al., 2017). Notably, ASVs affiliated with the Desulfobacterota groups *Desulfatiglans*, *Desulfobacteraceae*, and Seep-SRB4 were the 1[st] & 6[th], 15[th], and 17[th] most potentially relatively active (cDNA) ASVs in Clam Field seep sediments, respectively, and ASVs affiliated with the Chloroflexi were the 3[rd] and 8[th]

most potentially relatively active (Table S7).

      In seep sediments such as Clam Field where sulfate penetrates, sulfate reducers generally outcompete methanogens for hydrogen and acetate (Schönheit et al., 1982; Lovley and Klug, 1986; Reeburgh, 2007), leading to a local depletion in hydrogen and the possible stimulation of sulfate-dependent AOM (Hoehler et al., 1994; Lloyd et al., 2011; Kevorkian et al., 2021; Coon et al., 2023). However, certain methylated compounds like

methylamines and methyl-sulfides are more favorable for methylotrophic methanogens than sulfate reducers, allowing these methanogens to persist and be noncompetitive even in sediments where sulfate isn't fully depleted



(Oremland and Polcin, 1982; Winfrey and Ward, 1983), and where sulfate reducers are abundant. Methylated compounds can also be the degradation products of complex organic compounds (Yancey and Somero, 1980; Oremland et al., 1982; Alcolombri et al., 2015), and are likely ubiquitous in Clam Field sediments. Among *mcrA*-
containing taxa, the dominance of methylotrophic methanogens in Clam Field seep sediments supports this possibility. *mcrA* ASV.2, clustering with methylotrophic methanogens *Methanohalophilus halophilus* and *Methanomethylovorans hollandica*, was the most abundant at all sampled sediment depths within the Clam Field seep area, particularly at the shallowest sediment depths.

While the presence of non-methane hydrocarbons may explain the presence of particular
microorganisms, it's difficult to explain the near-exclusion of ANME archaea. Compared to AOM, the oxidation of non-methane hydrocarbons provides a higher energy yield per molecule of sulfate reduced (Bowles et al., 2011; Adams et al., 2013), particularly when energy yields from AOM must be split between the two partner organisms. While coexistence between ANME archaea and other hydrocarbon oxidizers has been documented elsewhere (Orcutt et al., 2010; Vigneron et al., 2017), it is possible that in the particular conditions of Clam Field seep,
coexistence is not viable. The precise mechanism of ANME exclusion – whether it be competition for essential nutrients, competition for sulfate as an electron acceptor, or inhibition by an unknown environmental factor – remains unknown.

## 5. Conclusion

The scarcity of known anaerobic methanotrophic populations at Clam Field has implications for
estimated methane emissions from marine cold seeps. Our findings suggest a previously unknown sensitivity of ANME to certain biogeochemical conditions and highlight the potential for hydrocarbon seeps without this critical methane biofilter. At Clam Field, this may result in the escape of 17.8-18.5 mmol methane $m^{-2}$ $yr^{-1}$. While aerobic methanotrophs in the water column can also prevent the release of methane to the atmosphere, and can thus curb its climate-warming impact, aerobic methanotrophy above seeps is poorly constrained due to potential variations
in oxygen concentration, the depth of overlying water, total methane flux at the seep (which can affect the presence/size of methane bubbles), and microbial community response time to new methane inputs. While current estimates suggest that anaerobic methanotrophs in sediments oxidize 80% of methane before it reaches the seafloor (Reeburgh et al., 2007), our data demonstrates large site-to-site variations in methanotrophic efficiency that are not reflected by the seep's surface expression or benthic macrofauna. Especially as new hydrocarbon
seeps develop along continental margins worldwide due to warming bottom waters and dissociating methane

hydrate reserves (Phrampus and Hornback; Skarke et al., 2014; Davies et al., 2023), direct observations of methanotrophs and methanotrophy will be necessary to confirm subsurface oxidation.

## 6. Data availability

Nucleotide sequences from this study were deposited in the European Nucleotide Archive, project
number PRJEB72125. Full sample metadata, including accession numbers, are available at
https://figshare.com/s/f282c579110b9bc9af70.

## 7. Author contribution

AS and AD designed the experiments and sampling campaign; AS collected the samples. AS performed sequencing, geochemical measurements, and microscopy, and analyzed the resulting data. AS and AD interpreted
the data together. AS wrote the initial manuscript draft; AD reviewed and edited the manuscript.

## 8. Competing interests

The authors declare that they have no conflict of interest.

## 9. Acknowledgments

We thank the captain, crew, and science party of R/V Western Flyer MMV19, particularly Nicolette
Meyer and Sebastian Sudek, as well as the pilots and engineers of ROV Doc Ricketts, for facilitating sample collection. We thank Alexandra Worden for the opportunity to collect samples on this expedition. We thank the members of the Dekas Geomicrobiology Lab for discussions and feedback, especially Steffen Buessecker for conversations about methane measurements. Funding was provided by Stanford University, including a McGee/Levorsen Research Grant to AS and through support to the Stanford Geomicrobiology Shared
Laboratories Core Facility (RRID:SCR_025000).



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
