# Peer review of "Unexpected scarcity of ANME Archaea in hydrocarbon seeps within Monterey Bay"

_EGUsphere, 2024_

## Author Comment (AC1)

The authors comprehensively characterized geochemistry and microbial communities of two seep sites in Monterey Bay, and observed very little DNA and RNA belonging to anaerobic methanotrophic archaea (ANME), particularly at Clam Field. They further supported this with gene quantification, microscopy, and sediment incubations under varying methane partial pressures, and posited that ANME may be outcompeted by other taxa in the presence of more complex hydrocarbons. They concluded that such a surprising absence of ANME has the potential to revise estimates of methane flux into the hydrosphere.

This manuscript is well-written, and the data are presented and interpreted in a logical order. I particularly appreciate how the authors contextualized their results as they presented them. I recommend accepting this manuscript once a few (mostly minor) details are clarified in the text. Hopefully my comments below are useful.

We appreciate this referee's thorough review, and we are happy to address their revisions and concerns in another draft. Our direct responses to their line-by-line comments are below in blue, with proposed textual additions in green.

**Methods:**
Lines 187-188: What was the minimum 16S sequencing depth of any sample? Were any samples discarded due to low read depth?
The minimum number of reads recovered per 16S rRNA gene sample (DNA) was 7,377, while the minimum number of reads recovered per 16S rRNA sample (cDNA) was 71. We will report these numbers in a future manuscript draft. No samples were discarded after sequencing, however in 3 16S rRNA samples (cDNA), the target did not amplify (and was therefore not sequenced). This is already shown in Fig. 4 – where bars are blank – and in Table S4. We will also add a clarification to the caption of Fig. 4.

Line 192: Please specify if the published mcrA sequences were manually compiled, or cite a relevant source.
The published *mcrA* sequences were manually compiled. This will be specified in a second draft.

**Results**
Line 295-296: Re "while the relative abundance of Bacteroidota decreased with sediment depth (Fig. 4a)" Just from looking at Fig. 4a, Bacteroidota look somewhat consistent with depth here- please justify with statistics or remove.
The decrease is extremely subtle in the plot but is clear in the values. We can therefore add the values (for all groups) to the sentence: "The relative abundances of Campylobacteria and Chloroflexi tended to increase with sediment depth in a core (by an average of 141% and 422%, respectively), while the relative abundance of Bacteroidota decreased (by an average of 53.4%) with sediment depth (Fig. 4a)."

Was no cDNA recovered from three of the samples at Clam Field seep-edge (Fig. 4a)?
Yes, correct! This is stated in Table S4, but we will make the following textual note to avoid confusion in the methods section: "At this stage, 16S rRNA genes and *mcrA* genes were successfully amplified from all 45 sediment horizons, while 16S rRNA was successfully amplified from 42 of 45 sediment horizons and *mcrA* transcripts were successfully amplified from 11 of 45 sediment horizons (Table S4)." We will also add a clarification to the caption of Fig. 4.

Line 336 "Seep-SRB1, the group containing many known obligate ANME symbionts" Please provide a citation or two.
Thanks for pointing this out. We will provide the following four citations that provide evidence of ANME/Seep-SRB1 symbiosis: Knittel et al., 2003; Schreiber et al., 2010; Skennerton et al., 2017; Metcalfe et al., 2021.

Lines 356-358: If I'm understanding correctly, the two highly abundant rows corresponding to ANME-2c in Extrovert Cliff both represent one ASV that cannot be resolved between the two reference sequences (and so its relative abundance is split between the two of them). I think it would minimize confusion if this were described more explicitly (particularly in the figure legend).
Yes, you're understanding correctly! This sentence is already in the methods section: "Relative abundances of ASVs placed on internal tree nodes were divided among tip nodes associated with that internal node." However, we will add a similar explanation to the figure caption for clarity. The caption will now include the following text: "Heatmap values were calculated by adding the relative abundance of all ASVs assigned to each tip node by EPA-ng (v. 0.3.8). (ASVs assigned to internal nodes were evenly divided among all tip nodes associated with that internal node. See methods.)"

Line 431: Please briefly justify the comparison to US Atlantic Margin seeps in particular: were they simply the most convenient to compare because similar data were collected by the same research group?
Yes, the USAM comparison sites were convenient to compare to because they were collected by the same research group, and all sampling, DNA/RNA extraction, sequencing, and sequence processing methodologies were standardized. In a future draft, the sentence will read: "To directly compare Monterey Bay seep microbial communities with canonical seep communities, we compared them to those of four seeps along the U.S. Atlantic Margin (USAM), which were sampled and sequenced with the same methodologies."

**Discussion**
Lines 478-483: I see some similar phrasing between this and the third paragraph of the introduction of the Semler et al 2022 AEM study. This feels borderline unnecessarily picky to point out, given the same first author, but I mention it just in case the editor or publisher disagrees with me.

Thank you for mentioning this. We will reword this statement in a future draft, so it conveys the same meaning but reads: "This community is primarily composed of ANME archaea (including ANME-1a, -ab; ANME-2a, -2b, -2c, -2d; and ANME-3) and SRB (including members of the Seep-SRB1 and Seep-SRB2 in the Desulfobacterales, *Desulfobulbus* and Seep-SRB4 in the Desulfobulbales; and thermophilic HotSeep-1). Sulfide-oxidizing and aerobic methane-oxidizing Gammaproteobacteria, as well as the putatively methanotrophic JS1 lineage of Atribacterota, are also abundant at seeps."

I understand that the lack of methane-dependent sulfate reduction and inability to enrich ANME in the clam field incubations would be considered a "negative" result, but why not comment on the Desulfobacterota that increased over time (Fig. S5?) Could these be the hydrocarbon degraders implicated in 553-555, and/or are they particularly good at thriving in sulfidic conditions?

Given the lack of methane-dependent sulfide production, and given that we did not add any non-methane hydrocarbons to our incubation headspace, we believe that the sulfate reducers enriched in these incubations do not make a living off of hydrocarbon degradation – they are instead thriving under increasingly sulfidic conditions. However, we will clarify in the text that the comparisons displayed in Fig. S5 were only comparisons between timepoints, not between methane headspace treatments; comparisons of T=6 WITH methane vs. T=6 WITHOUT methane yielded no significant taxa enrichments or un-enrichments, indicating that while communities changed over time, the presence of methane did not seem to cause these differences.

---

## Author Comment (AC2)

The manuscript by Semler & Dekas describes the biochemistry and microbial ecology of two cold seep sites located in the Monterey Bay area. The authors report low abundances of ANME archaea quantified by 16S rRNA gene and transcript sequencing, together with ddPCR and mcrA sequencing, complemented by microscopic identification of typical ANME-SRB aggregates, low rates of potential activity and lack of response to the presence of methane during prolonged incubations. These microbial ecology observations are complemented by geochemical measurements of relevant compounds (methane, sulfide, sulfate) and isotopic measurements of methane.

The manuscript is well written and tackles an interesting question. The data seem to be carefully collected and are described and analyzed with scientific rigor. Conclusions follow environmental observations or experimental outcomes and hypotheses are presented as such in case they could not be confirmed by experimentation/observation. My comments are all minor. I congratulate the authors for this very interesting piece of work

We appreciate this referee's positive comments about the overall craftsmanship, and we are happy to address their revisions and concerns in another draft. Our direct responses to overall and line-by-line comments are below in blue, with proposed textual additions in green.

While the DAPI staining nicely shows aggregates of cells at Extrovert Cliff, their taxonomic identity is unclear. The authors are aware of this (L399) and describe them as putative ANME aggregates (Fig S2 caption) and claim that an ANME-typical morphology was detected (L508). Would it be possible to explain this in the discussion? Is this morphology really typical? Is there any previous experience with such an indirect classification? This would help the reader to see the validity of the experimental strategy.

Yes, there is some precedent for this kind of indirect classification, and we will add the following sentences to the results section to explain our thinking: "ANME-2 and ANME-3 typically form tight associations with their syntrophic partners, resulting in cellular aggregates of characteristic morphologies (Boetius et al., 2001; Orphan et al., 2022). While not a taxa-specific assay, quantifying DAPI-stained cell aggregates typical of the ANME-SRB morphology provides independent support for the presence or absence of aggregate-forming ANME-2 and ANME-3 archaea in the molecular data, and has been used previously to approximate potential ANME-SRB aggregate abundances (Dekas et al., 2009; Zhang et al., 2011)."

My second comment relates to the extended incubation. I wonder why only the Clam Field site was chosen for incubation, even though AOM for the other site was more likely but still low compared to other cold seep sites. I suggest stating the reason in one

sentence. Direct observation/quantification of sulphate-dependent AOM at Extrovert Cliff would allow comparison with other sites.

We set up incubations from Clam Field immediately after the sampling trip – before performing any sequencing on either site. It was chosen mainly based on the surface expression of the seep (patchy mats and clams), the sulfidic smell of sediments, and its similarity to others that had been sampled previously. We hadn't anticipated the absence of ANME at either site. We will add the following sentence to our "Incubation setup" section in the methods to address this comment: "Clam Field was chosen for incubations because its surface expression was similar to that of seeps sampled and characterized in previous studies (McVeigh et al., 2018; Seabrook et al., 2018; Semler et al., 2022), and thus responses of characteristic seep microbial communities to varying methane headspace concentration could be tested."

Minor comments:
L19 - I feel that the last sentence of an abstract needs polishing.

The current text reads: "Our findings highlight the potential for hydrocarbon seeps without this critical biofilter, with implications for their contribution to global methane emissions." We suggest the following change: "Our findings highlight the potential for hydrocarbon seeps without this critical biofilter, and therefore unabated methane emission."

L56 – please consider to include the new clade names for (doi: 10.1371/journal.pbio.3001508.)

We will introduce these new clade names in the introduction at L56 – thank you for the suggestion!

L58 and elsewhere – you use 'symbiont' for the ANME-SRB aggregates while I find 'syntrophs' much more fitting, as it describes the type of metabolic interaction

We will make this word substitution throughout the manuscript.

L183 - Was it possible to overlap majority of pair-end reads after trimming? What was the length of overlap in DADA2 pipeline?

Yes, the majority of 16S rRNA paired-ends were merged. The following sentence will be added to the methods section: "The majority of paired-ends were merged for both genes (an average of 76% for 16S rRNA reads in non-incubated samples and 95% of *mcrA* reads), and the overlap length was roughly 30 bp for 16S rRNA and 50 bp for *mcrA*."

L188 - 53K mcrA reads per sample are not in line with 39K reads reported in the Results (L352), please check.

Thank you for noticing this – one value referred to *mcrA* genes AND transcripts, while the other value referred to genes only (and one of the values was mistakenly

calculated). We will make the distinction clearer in both sections and will ensure the values are correct.

L195 - Please specify the substitution model which was used to construct the reference tree.
The substitution method was GTR+G+I. This will be added to the methods section.

L393 - Converting mcrA copy number per well to cells per g assumes mcrA gene to be a single copy gene which is not true in 100% cases, cell numbers can be overestimated.
This is a great point that we can address in a second draft of this manuscript. Happily, most methanogens (and all ANME that have available genomes) appear to have only one or two copies of *mcrA*. We therefore propose the following textual addition: "The assumption of one *mcrA* copy per cell is imperfect, but only one or two copies of mcrA have been found in sequenced methanogen genomes (Alvarado et al., 2014), including that of *Methanosarcina mazei* – a close relative of ANME-2 (Deppenmeier et al., 2002; Nunoura et al., 2009). The ANME-1 genome also contains a single operon for MCR (Krukenberg et al., 2018; Chadwick et al., 2022; Laso-Pérez et al., 2023). As a result, the roughly 4 order of magnitude difference in ANME cell numbers between Clam Field and Extrovert Cliff would not be significantly affected by likely variation in ANME *mcrA* copy number."

L289 and elsewhere – please use 16S rRNA gene amplification rather than 16S rRNA amplification
Thank you for pointing this out. We will change the section title to: "Community composition at Monterey Bay cold seeps via 16S rRNA gene and 16S rRNA amplicon sequencing", and we will specify 16S rRNA gene and 16S rRNA on L290 as well.

L306 -as inferred by the presence of…
Will be corrected in a future draft.

L429 - Methods section is missing the description of combining ASV matrices from the current and previous studies, please add such description.
16S rRNA gene and 16S rRNA samples from all sites (Monterey Bay and USAM) were actually processed together – the raw USAM data was processed with the raw Monterey Bay data before analysis – and thus have the same ASV matrices. We will put the following sentence in the methods section 2.8: "Raw data from four U.S. Atlantic Margin (USAM) seep sites (characterized in Semler et al. 2022) was simultaneously processed using the same packages for comparison with Monterey Bay sites."

L605 - ENA accession number is still private, please either release the dataset or share a reviewer's link.
The data should now be released at the accession numbers we specified in Section 6. (There may be a delay until 9/28.)